# Production and Characterization of Nanostructured Powders of Nd_2_Fe_14_B and Fe_90_Al_10_ by Mechanical Alloying

**DOI:** 10.3390/molecules27217190

**Published:** 2022-10-24

**Authors:** Alvaro Javier Gómez Rodríguez, Dagoberto Oyola Lozano, Humberto Bustos Rodríguez, Yebrail Rojas Martínez, German Antonio Pérez Alcázar, Ligia Edith Zamora Alfonso, Juan Sebastian Trujillo Hernandez

**Affiliations:** 1Departamento de Física, Universidad del Tolima, Ibagué 730006299, Colombia; 2Grupo de Metalurgia Física y Teoría de Transiciones de Fase, Departamento de Física, Universidad del Valle, Cali 25360, Colombia; 3Centro de Excelencia de Nuevos Materiales (CENM), Universidad del Valle, Cali 25360, Colombia

**Keywords:** magnetic materials, Mössbauer spectrometry, mechanical alloying, nanoparticle

## Abstract

The objective of this work is to evaluate the applicability of exchange coupling between nanoparticles of Nd_2_Fe_14_B (hard magnetic material) and Fe_90_Al_10_ (soft magnetic material), as permanent magnets produced by surfactant-assisted mechanical alloying. The obtained powders were then mixed with 85% of the Nd_2_Fe_14_B system and 15% of the Fe_90_Al_10_ system and subsequently sintered at 300 °C, 400 °C and 500 °C for one hour. The results obtained by Mössbauer spectrometry (MS) show a ferromagnetic behavior with six magnetic sites represented by sextets (16k1, 16k2, 8j1, 8j2, 4c and 4e), characteristic of the Nd_2_Fe_14_B system. X-ray diffraction (XRD) results show a tetragonal and BCC structure for the Nd_2_Fe_14_B and FeAl systems, respectively. The results obtained by vibrating sample magnetometry (VSM), for mixtures of the Nd_2_Fe_14_B and Fe_90_Al_10_ sy stems sintered at 300 °C, 400 °C and 500 °C, allow for the conclusion that the coercive field (Hc) decreases drastically with temperature and the percentage of soft phase at values of Hc = 132 Oe compared to the coercive field values reported for Nd_2_Fe_14_B Hc = 6883 Oe, respectively. Images obtained by transmission electron microscopy (TEM), for the Fe_90_Al_10_ system, show a tendency for the nanoparticles to agglomerate.

## 1. Introduction

Permanent magnets based on rare earth (RE) elements, such as Nd_2_Fe_14_B, play a fundamental role in current and emerging technologies; they are found in commonly used devices (computers, speakers, smartphones, etc.), in the development of electric vehicles, and are an indispensable input for the construction of electrical generators capable of producing electrical energy from renewable natural sources such as water (hydroelectric power), wind (wind power) and the sea (wave energy) [1,2,3,4,5,6,7]. For this reason, the design of high-performance permanent magnets represents a need at the industrial level [1,2,5,6,7]. Permanent magnets usually contain large amounts of RE and other elements that are difficult to supply, such as Pr, Sm, Dy, Tb, Nd and Co, most of which are mined in foreign countries such as China, Congo, Russia and the United States [1,2,3,4]. Therefore, a huge gap crisis has been generated between the supply and demand of these magnets due to the unavailability of these raw materials and the difficulty in supplying them [1,2,3]. Taking into account this problem, recent studies have focused on the search for alternatives to produce magnets with a high energy density (BH)max using the smallest possible amount of rare earths [1,2,5,7,8,9,10]. Thus, the development of producing a two-phase nanocomposite formed by two magnetic phases, a hard one with high anisotropy (NdFeB or SmCo) coupled with a soft one with high magnetization (Fe, FeSi or FeCo), has gained great relevance in recent years and is considered the new generation of permanent magnets [1,4,7,8,9,10,11]. These exchange-coupled magnets average the magnetization and anisotropy of the two constituents [8]. However, to have a positive impact on (BH)max, one of the direct effects is the improvement of the coercivity by the reduction of the grain size, which should be proportional to the width of the hard phase domain wall, allowing for the optimization of the exchange coupling [1,2,3,4,8]. The great challenge for researchers today remains in designing nanocomposites at a nanometer scale where the separation between hard and soft grains does not exceed 50 nm [1,2,3,4,8]. Recently, it has been shown that one process route to obtain nanoparticles with grain sizes smaller than 30 nm is by surfactant-assisted mechanical alloying [12]. In this work, Nd_2_Fe_14_B and Fe_90_Al_10_ samples produced by arc melting and surfactant-assisted mechanical alloying were structurally and magnetically characterized. In addition, the exchange coupling study of a nanocomposite formed by Nd_2_Fe_14_B/Fe_90_Al_10_ and heat treated at different temperatures (300 °C, 400 °C and 500 °C) for one hour was carried out.

## 2. Results

### 2.1. Fe_90_Al_10_ System

Figure 1 shows the X-ray diffraction patterns of the Fe_90_Al_10_ arc furnace melt powders, sintered at 1000 °C for 7 days, and subsequently mechanically alloyed with oleic acid as a surfactant for 24 and 48 h. Here, we can observe that for the three diffractograms, there is only the presence of the phase corresponding to the Fe_90_Al_10_ system, with a BCC (cubic) structure and lattice parameter of (2.88 Å) [13,14,15]. This is a consequence of the effect of the milling time by not promoting secondary phases. Additionally, as shown in Table 1, we can observe the X-ray parameters (lattice parameter, crystallite size and weight fraction) of the crystalline phases present in the diffractograms of Figure 1, which were refined with an FeAl phase.

Figure 2 shows the TEM images of the Fe_90_Al_10_ nanoparticles obtained by surfactant-assisted mechanical alloying after 48 h. It can be observed that the milling time reduces the particle size with an irregular shape with an average width in the range of 30–70 nm and a length of 80–250 nm after 48 h of milling time, promoting agglomeration [12].

Figure 3 shows the Mössbauer spectra and the hyperfine field probability distributions of the arc furnace powders of the Fe_90_Al_10_ system, sintered at 1000 °C for 7 days. The Mössbauer spectra were fitted with one component (sextet) associated with the Fe_90_Al_10_ system, indicating a ferromagnetic behavior which presents a disorderly character given by the half-width of the line due to the reduction of the particle size by the milling process [13,15]. This is a consequence of the high content of Fe in the sample and the thermal treatment carried out on it at 1000 °C followed by quenching in ice water, and this treatment retains the high-temperature disordered structure. Due to the line width, the spectra of the sample was fitted with a magnetic hyperfine field distribution (HMFD). The result of the HMFD is shown on the right side of Figure 3, where it is observed that the distribution has a single site, or more likely a site near 310 kOe. After fitting, a mean hyperfine field of 318 kOe is obtained which is smaller than the field of pure Fe as a consequence of the substitution of Fe by Al atoms. These results are similar to those reported by other authors [13].

Figure 4 shows the spectra of the sample mechanically alloyed with oleic acid as a surfactant for 48 h of milling time, which was fitted with an HMFD (blue line), whose spectral area corresponds to 70%, with a mean-field of 320 kOe, and a doublet whose spectral area is 30% (red line) [13]. This behavior can be explained because the mechanical alloying promotes the diffusion of atoms in the BCC structure. Therefore, now 70% of the Fe sites will be surrounded by many first neighbors of Fe, leading to ferromagnetic behavior, and 30% of the remaining Fe sites will be surrounded by first neighbors of Al, leading to paramagnetic behavior. This interpretation is due to the unique presence of the BCC phase, as given by XRD. Here, the average hyperfine field of the milled sample is higher than the average hyperfine field of the thermally treated sample, indicating that the ferromagnetic sites of the milled sample are richer in Fe than the sites of the thermally treated sample (fewer atoms of the first neighbors). Mössbauer parameters are shown in Table 2.

Figure 5 shows the hysteresis loop of the Fe_90_Al_10_ system and Table 3 shows the values of the extrinsic magnetic properties of the material: saturation magnetization (Ms), remanent magnetization (Mr) and coercive field (Hc) obtained from the hysteresis loops. These results indicate that the samples have ferromagnetic behavior (soft magnetic material), according to the results of the Mössbauer spectra. Additionally, it can be seen that all magnetic properties decrease as the milling time increases. This is due to the reduction in particle size, which causes the diffusion of Fe atoms and promotes paramagnetic sites in the system, as observed in Mössbauer, possibly due to the appearance of paramagnetic sites, as Mössbauer showed.

### 2.2. Nd_2_Fe_14_B System

Figure 6 shows the X-ray diffraction patterns of the commercial powder of the Nd_2_Fe_14_B system and the powder produced by mechanical alloying assisted with oleic acid as a surfactant for 24 and 48 h of milling time. The structural parameters (lattice parameter, crystallite size and fraction of volume) of the crystalline phases present in the diffractograms of Figure 6 are shown in Table 4. Here, we can observe that before and after milling time, the powders have a tetragonal structure associated with the Nd_2_Fe_14_B phase (hard phase) [1,4]. However, with the increase in the milling time, we can observe an increase in the intensity of the peak occupied by the position (2θ = 26.5°, this peak was not possible to associate with any known phase). Furthermore, it is observed that the intensity of the peaks of the hard phase decreases and the half-width of the line increases, indicating a reduction of the crystallite size. The XRD parameters obtained after the Rietveld refinement indicate that the perpendicular and parallel crystallite sizes, and also the lattice parameters, decrease with the milling time. This result indicates that the milling process is effective in reducing the size of crystallites and increasing micro-tensions, which explains why the half-width of the line of the diffractogram increases. In addition, surfactant-assisted mechanical alloying promotes particle size reduction.

The Mössbauer spectrum of the powders of the Nd_2_Fe_14_B system mechanically alloyed with oleic acid as a surfactant for 48 h of milling time is shown in Figure 7. The Mössbauer parameters of the sample are presented in Table 5. The sample was fitted with six magnetic sites (sextets) associated with six non-equivalent crystallographic iron sites corresponding to the hard magnetic phase Nd_2_Fe_14_B (16K_1_, 16K_2_, 8j_1_, 8j_2_, 4c, and 4e); which indicates that the sample presents a ferromagnetic behavior [16,17].

Figure 8 shows the hysteresis loops of the commercial powder of the Nd_2_Fe_14_B system and the powder obtained by the milling process with surfactant for 24 and 48 h at room temperature. Table 6 shows the values of the extrinsic magnetic properties: saturation magnetization (Ms), remanent magnetization (Mr) and coercive field (Hc) obtained from the hysteresis loops. The result showed the characteristic behavior of a single-phase hard magnet with no kinks in the demagnetization curve, with coercive field values in the range of 2350.9 to 6883.4 Oe, saturation magnetization values in the range of 116 to 129 emu/g, and values of remanent magnetization values in the range of 48.4 to 84.5 emu/g. The highest coercivity of 6883.4 Oe was obtained in the pure sample. Coercivity values decrease with the decrease in particle size, which means that particles in the nanometric order will lead to lower coercivity, although the lowest coercivity range occurs at 24 h of milling time. The same decrease occurs for the remanent magnetization and saturation magnetization when the particle size decreases.

### 2.3. Nd_2_Fe_14_B/Fe_90_Al_10_ Nanocomposite System

Figure 9 shows the X-ray diffraction patterns of the Nd_2_Fe_14_B/Fe_90_Al_10_ system thermally treated at 300 °C, 400 °C and 500 °C. The X-ray parameters (lattice parameter, crystallite size and volume fraction) of the crystalline phases present in the diffractograms of Figure 9 are shown in Table 7. Here, it is observed that for all temperatures that the BCC structure of the Fe_90_Al_10_ system [13] predominates and the tetragonal structure of the Nd_2_Fe_14_B system disappears almost completely. The parameters obtained by Rietveld refinement indicate that the majority phase is Fe_90_Al_10_ with a volume fraction of almost 1.0, confirming the absence of the tetragonal phase of Nd_2_Fe_14_B in the mixing samples. This is possibly due to the size of the crystallites of the Fe_90_Al_10_ system, and/or that the soft phase mass ratio is excessive, which makes it possible to identify only this phase. The perpendicular and parallel crystallite sizes increase for the Fe_90_Al_10_ system and show a significant reduction for the Nd_2_Fe_14_B system. The lattice parameters remain constant for the Fe_90_Al_10_ system, while for the Nd_2_Fe_14_B phase they have an irregular behavior; however, the low intensity of this phase in the diffractogram can cause errors in the measurement of these values. The Mössbauer spectra of the Nd_2_Fe_14_B/Fe_90_Al_10_ system sintered for one hour at 300 °C are shown in Figure 10. The Mössbauer parameters for each sample are presented in Table 8. The sample was fitted with a sextet and a singlet corresponding to the Fe_90_Al_10_ phase [16]. The spectra corresponding to the tetragonal phase of Nd_2_Fe_14_B do not appear, which effectively shows the absence of this phase in the mixture samples as observed by XRD.

Figure 11 shows the hysteresis loops of the commercial powders of the Nd_2_Fe_14_B system and the powders obtained by mechanical alloying assisted with surfactant for 24 and 48 h of milling time at room temperature. Table 9 shows the values of the extrinsic magnetic properties: saturation magnetization (Ms), remanent magnetization (Mr) and coercive field (Hc) corresponding to the hysteresis loops. The hysteresis loops show a characteristic behavior of a single-phase soft magnet with no kinks in the demagnetization curve, with coercive field values in the range of 132 to 171 Oe, saturation magnetization values in the range of 138 to 173 emu/g, and remanent magnetization values in the range of 9 to 16 emu/g. The highest coercivity of 171 Oe was obtained in the sample sintered at 300 °C. As seen in Figure 11, the magnetic hardness offered by the tetragonal Nd_2_Fe_14_B phase is completely lost. Additionally, the increase in the sintering temperature decreases the magnetic properties, such as saturation magnetization (Ms), remanent magnetization (Mr) and coercive field (Hc).

## 3. Experimental Process

Fine Fe and Al powders of 99.5% and 99% purity were melted in an arc furnace and sintered at 1000 °C for seven (7) days. From the alloy obtained, powders were obtained with a diamond file. These powders and the fine powder of Nd_2_Fe_14_B (with a purity of 99% and an average particle size of 150) were mechanically alloyed with surfactant (oleic acid) in a planetary ball mill (Fritsch Pulverisette 5) at 280 rpm with a ball-to-powder ratio of 20/1 for 24 and 48 h milling times [18]. Finally, the soft (Fe_90_Al_10_) and hard (Nd_2_Fe_14_B) phases were mixed for 48 h, with milling times at 15% and 85%, respectively. Then, the mixture was heat treated at 300 °C, 400 °C and 500 °C for one hour. Structural and magnetic characterization were performed by X-ray diffraction (XRD), transmission electron microscopy (TEM), Mössbauer spectrometry (MS) and vibrating sample magnetometry (VSM). XRD measurements were performed at room temperature using Cu-Kα radiation. The diffraction patterns were refined using the Rietveld method using the GSAS program [19], from the refinement of the calibration sample (LaB_6_). The images obtained by TEM show the morphology and size of the Nd_2_Fe_14_B nanoparticles. The Mössbauer spectra were fitted using Mosfit software [20], and were performed at room temperature using a ^57^Co (Rh) source with transmission geometry. The VSM measurements were obtained at room temperature, with an applied field up to 3 T, using a physical properties measurement system (PPMS) without taking demagnetization effects into consideration.

## 4. Conclusions

The Fe_90_Al_10_ alloy produced by surfactant-assisted mechanical alloying forms a soft magnetic material with a BCC structure. The milling process decreases the particle size to the nanometer scale, promoting a tendency to agglomeration [21]. The Mössbauer results showed that the hyperfine field of the material decreases with milling time, showing a paramagnetic site due to the substitution of aluminum atoms by iron atoms in its crystal structure. The commercial Nd_2_Fe_14_B alloy presents a tetragonal structure; this was mechanically alloyed, and assisted with surfactant in order to reduce the particle size. The Mössbauer results showed the presence of six magnetic sites (sextets), characteristic of the Nd_2_Fe_14_B unit cell [1,2,3,4,5]. The VSM results show a reduction of the coercive field with milling time, which is related to the increase of the amorphous part in the sample as a consequence of the milling time. The nanocomposite formed by (Nd_2_Fe_14_B/Fe_90_Al_10_), and heat treated at 300, 400 and 500 °C, does not show an exchange coupling between the two magnetic systems, evidencing a soft magnetic behavior, characteristic of the Fe_90_Al_10_ alloy, as observed by XRD.

## Figures and Tables

**Figure 1 molecules-27-07190-f001:**
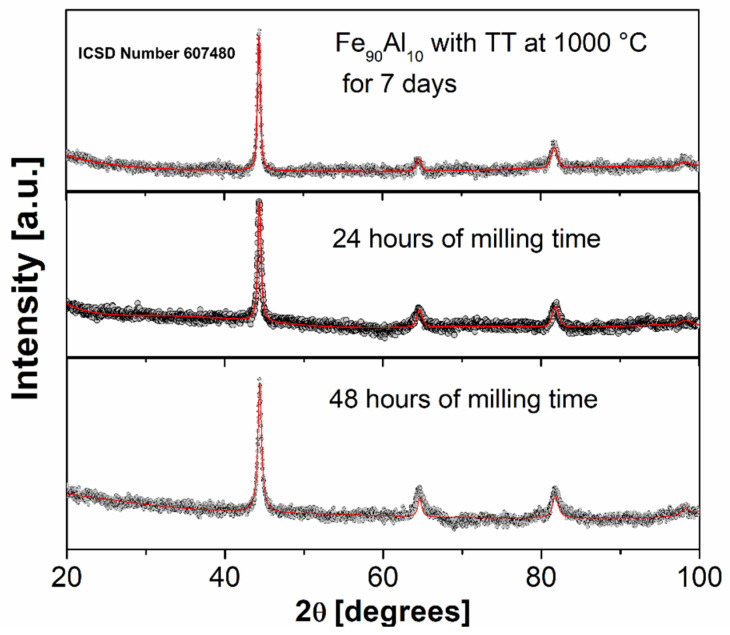
XRD of the Fe_90_Al_10_ system with thermal treatment (TT) at 1000 °C for 7 days and mechanically alloyed with oleic acid as a surfactant for 24 and 48 h, respectively.

**Figure 2 molecules-27-07190-f002:**
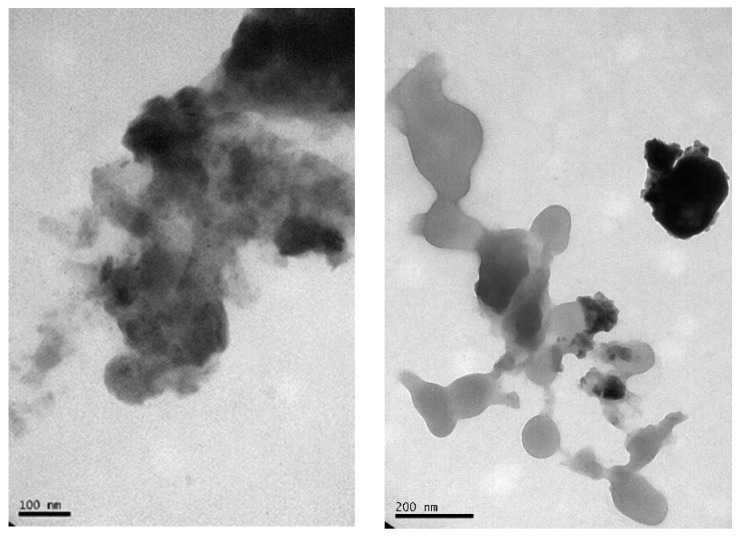
Transmission electron microscopy (TEM) images of the Fe_90_Al_10_ system with 48 h of mechanical alloying with surfactant.

**Figure 3 molecules-27-07190-f003:**
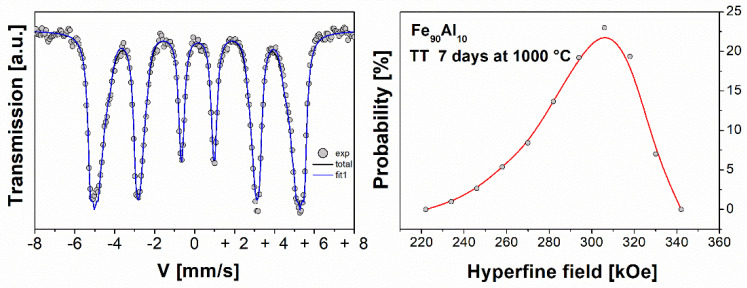
Mössbauer spectra and its corresponding probability distribution of the Fe_90_Al_10_ system with thermal treatment (TT) at 1000 °C for 7 days.

**Figure 4 molecules-27-07190-f004:**
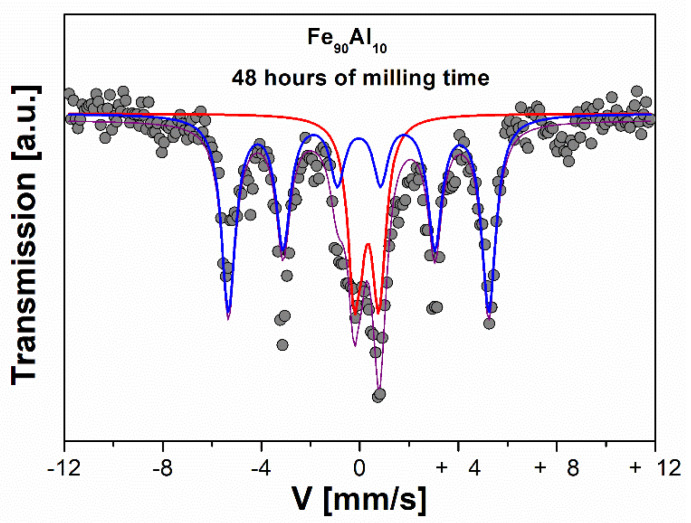
Mössbauer spectra of the Fe_90_Al_10_ system with thermal treatment at 1000 °C for 7 days and milled for 48 h with oleic acid as a surfactant. The fit of the spectrum is given by the theoretical line (purple line), the HMFD (blue line) and the doublet (red line).

**Figure 5 molecules-27-07190-f005:**
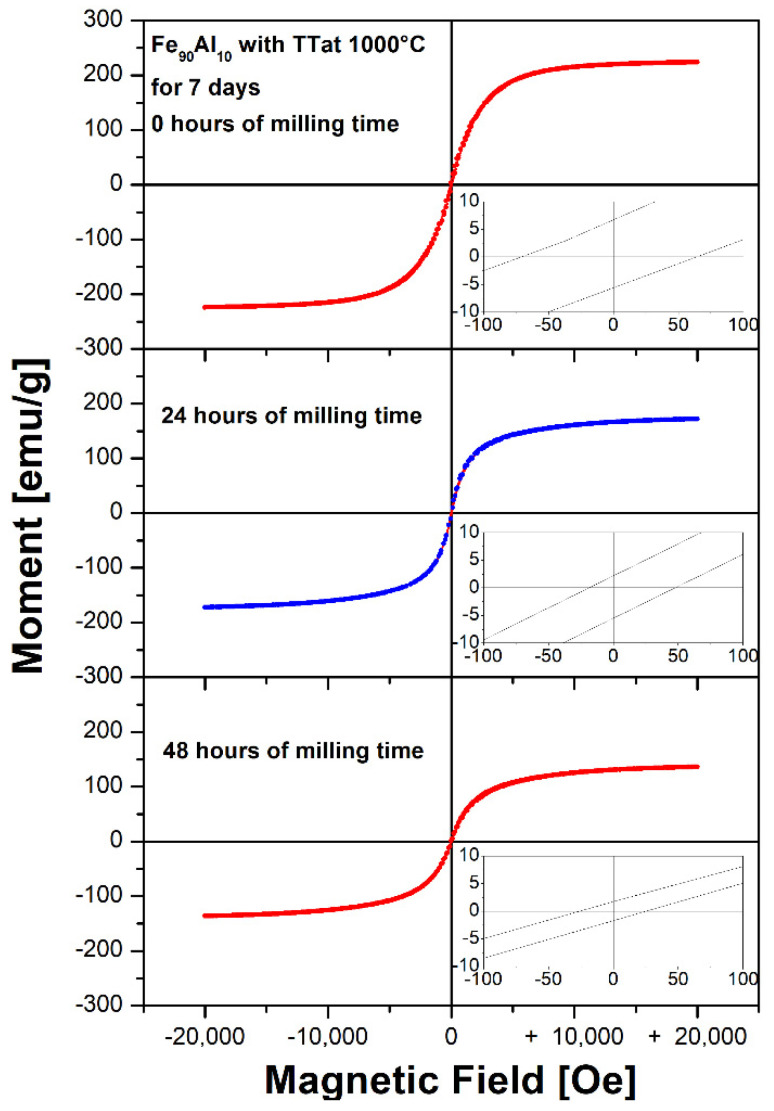
Hysteresis loops Fe_90_Al_10_ system with thermal treatment at 1000 °C for 7 days and mechanically alloyed for 24 and 48 h, respectively.

**Figure 6 molecules-27-07190-f006:**
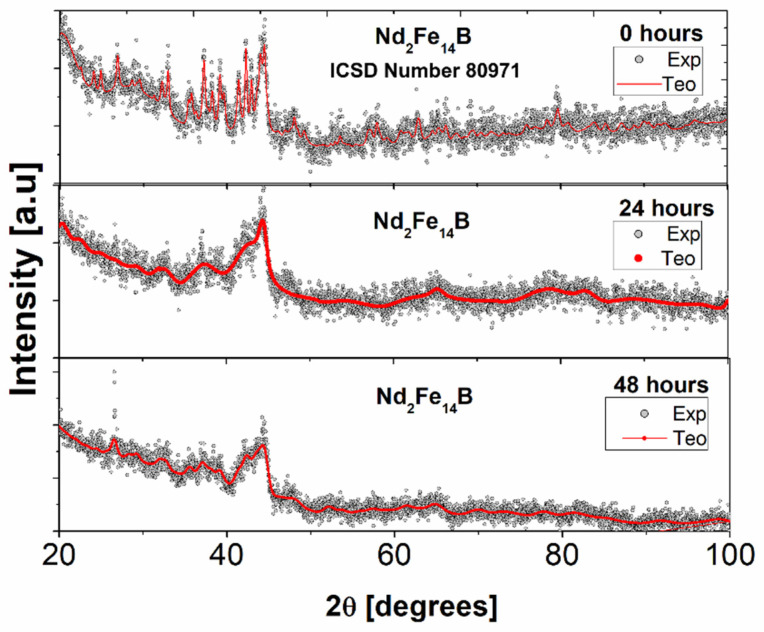
XRD of the Nd_2_Fe_14_B system without milling and mechanically milled with oleic acid as a surfactant for 24 and 48 h, respectively.

**Figure 7 molecules-27-07190-f007:**
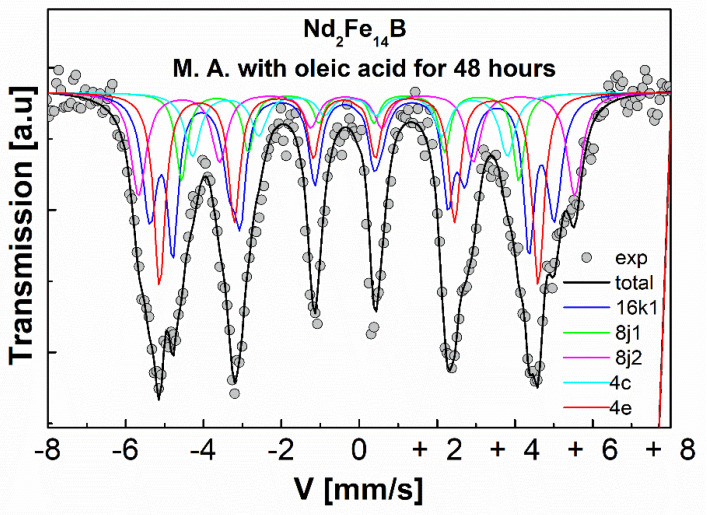
Mössbauer spectra of the Nd_2_Fe_14_B system mechanically alloyed (M.A.), with oleic acid for 48 h.

**Figure 8 molecules-27-07190-f008:**
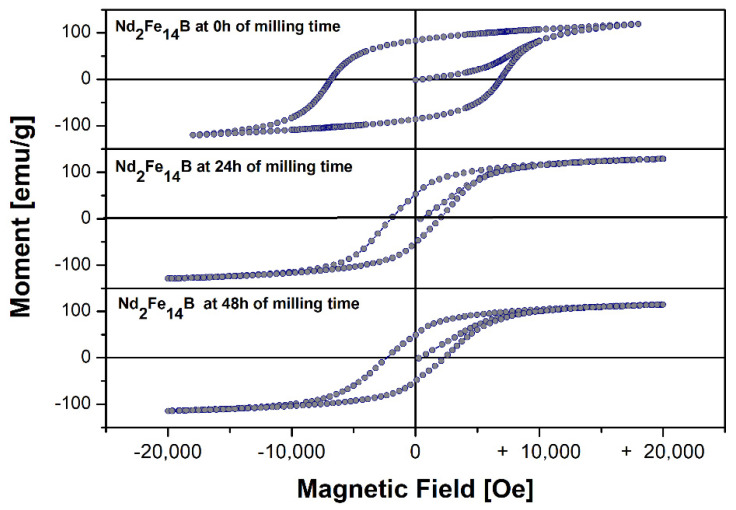
Hysteresis loops of the Nd_2_Fe_14_B system and mechanically alloyed for 24 and 48 h, respectively.

**Figure 9 molecules-27-07190-f009:**
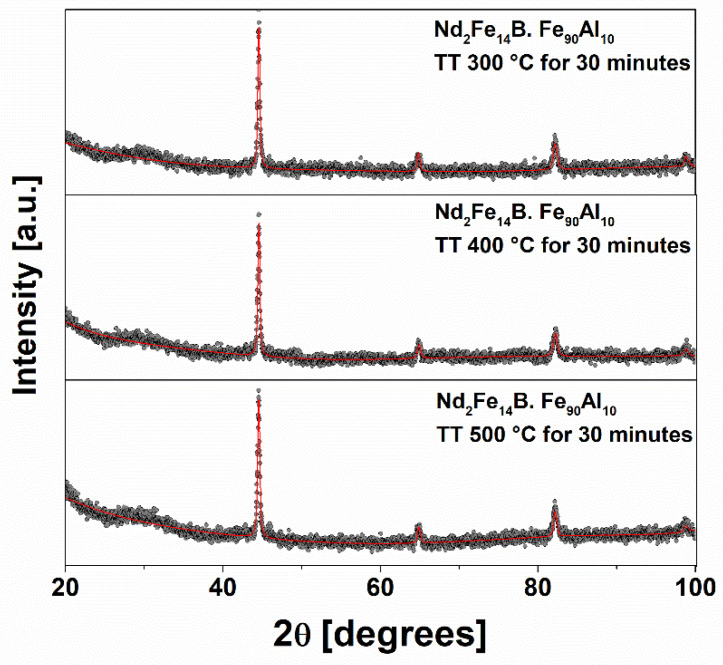
XRD of the sintered Nd_2_Fe_14_B/Fe_90_Al_10_ systems at 300, 400 and 500 °C, respectively.

**Figure 10 molecules-27-07190-f010:**
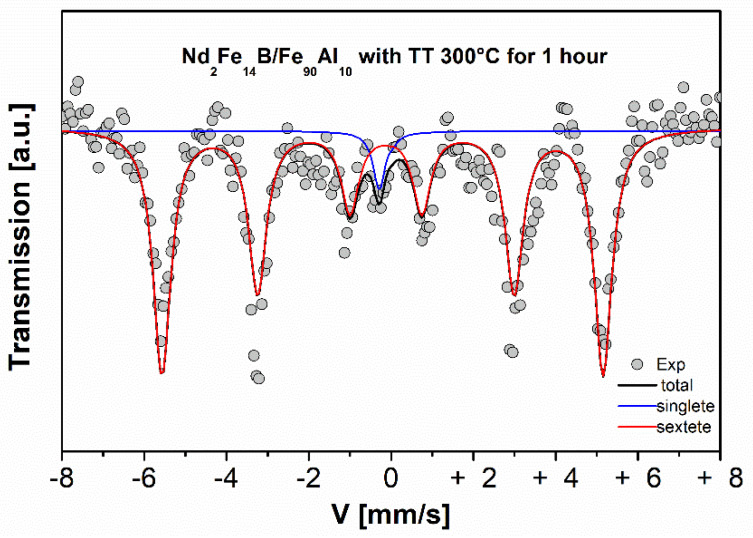
Mössbauer spectra of the sintered Nd_2_Fe_14_B/Fe_90_Al_10_ system for one hour at 300 °C.

**Figure 11 molecules-27-07190-f011:**
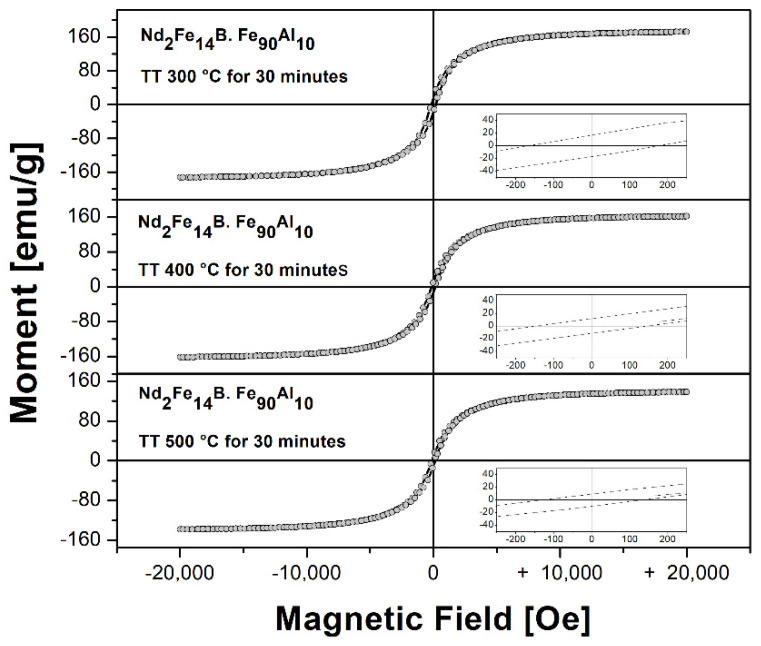
Hysteresis loops of the sintered Nd_2_Fe_14_B/Fe_90_Al_10_ system at 300, 400 and 500 °C, respectively.

**Table 1 molecules-27-07190-t001:** XRD parameters of the Fe_90_Al_10_ system with thermal treatment at 1000 °C for 7 days and mechanically alloyed with oleic acid as a surfactant for 24 and 48 h of milling time, respectively (the (a) column corresponds to the lattice parameters; the columns with (Φ_per_) correspond to the mean crystallite size perpendicular to the radiation, the column with (Φ_par_) corresponds to the mean crystallite size parallel to the radiation).

Time	Density (g/cm^3^) ±0.04	Volume (Å^3^) ±0.04	a (Å) ±0.04	Φ_per_ (nm) ±0.5	Φ_par_ (nm) ±0.5
0 h	6.96	24.041	2.88	28.8	14.4
24 h	6.99	23.937	2.88	21.7	18.1
48 h	6.97	23.990	2.88	24.8	12.4

**Table 2 molecules-27-07190-t002:** Mössbauer parameters and probability distribution of the Fe_90_Al_10_ system with thermal treatment at 1000 °C for 7 days and 48 h of milling with oleic acid as a surfactant (the column with (δ) corresponds to the isomer shift values; the column with (Γ) corresponds to the line widths; the column with (*Qs*) corresponds to the quadrupole splitting; the column with (HF) corresponds to the hyperfine field; and the line with (HMDF) corresponds to the magnetic hyperfine field distribution).

Time	Component	δ (mm/s) ±0.005	Г (mm/s) ±0.005	*Qs* (mm/s) ±0.005	HF (kOe) ±0.4	Spectral Area %
0 h	HMDF	0.033		−0.026	318.2	
48 h	HMDF Doublet	0.038 0.383	0.535	0.323 0.888	200.1	62.5 37.5

**Table 3 molecules-27-07190-t003:** Extrinsic magnetic properties of the Fe_90_Al_10_ system with thermal treatment at 1000 °C for 7 days and milled for 24 and 48 h with oleic acid as a surfactant (the column with (Hc) corresponds to the coercive field; the column with (Mr) corresponds to the remanent magnetization; and the column with (Ms) corresponds to the saturation magnetization).

Milling Time	Hc (Oe) ±0.4	Mr (emu/g) ±0.4	Ms (emu/g) ±0.4
0 h	67.5	6.8	225.2
24 h	33.3	2.3	173.7
48 h	26.6	1.7	137.9

**Table 4 molecules-27-07190-t004:** Structural parameters of the original system Nd_2_Fe_14_B and mechanically alloyed with oleic acid as a surfactant for 24 and 48 h, respectively (the columns (a and c) correspond to the lattice parameters).

Milling Time	Density (g/cm^3^) ±0.04	Volume (Å^3^) ±0.04	a (Å) ±0.04	c (Å) ±0.04	Φ_per_ (nm) ±0.5	Φ_par_ (nm) ±0.5
0 h	7.58	947.26	8.80	12.20	54.7	34.4
24 h	8.25	931.63	8.75	12.14	8.0	15.2
48 h	9.09	899.08	8.63	12.04	14.8	22.0

**Table 5 molecules-27-07190-t005:** Mössbauer parameters of the commercial Nd_2_Fe_14_B system for 48 h of milling time with oleic acid as a surfactant. The column with (HF) corresponds to the hyperfine field; and the lines with (J1, J2, K1, K2, C, E) correspond to the non-equivalent crystallographic iron site for Nd_2_Fe_14_B).

Milling Time	Component	δ (mm/s) ±0.005	Г (mm/s) ±0.005	*Qs* (mm/s) ±0.005	HF (kOe) ±0.4	Spectral Area (%)
48 h	J1	−0.074	0.165	0.171	282.3	17.4
	J2	−0.017	0.231	0.112	329.1	19.6
	K1	−0.054	0.174	0.097	267.1	10.8
	K2	0.026	0.234	0.263	346.4	17.0
	C	−0.006	0.239	0.114	249.2	10.5
	E	−0.097	0.181	0.114	300.4	24.5

**Table 6 molecules-27-07190-t006:** Extrinsic magnetic properties of the Nd_2_Fe_14_B system and milled for 24 and 48 h with oleic acid as surfactant.

Time	Hc (Oe) ±0.4	Mr (emu/g) ±0.4	Ms (emu/g) ±0.4
0 h	6883.4	84.5	118.4
24 h	1979.2	52.7	129.3
48 h	2350.9	48.3	115.9

**Table 7 molecules-27-07190-t007:** XRD parameters of the sintered Nd_2_Fe_14_B/Fe_90_Al_10_ system at 300, 400 and 500 °C, respectively.

Temperature (°C)	Phase	Density g/cm^3^ ±0.04	Volume (Å^3^) ±0.04	a (Å) ±0.04	Φ_per_ (nm) ±0.5	Φ_par_ (nm) ±0.5
300	Fe_90_Al_10_	5.23	23.39	2.85	47.2	20.0
400	Fe_90_Al_10_	7.04	29.75	2.87	62.6	34.1
500	Fe_90_Al_10_	7.07	23.67	2.87	43.9	43.2

**Table 8 molecules-27-07190-t008:** Mössbauer parameters of the sintered Nd_2_Fe_14_B/Fe_90_Al_10_ system for one hour at 300 °C.

Component	δ (mm/s) ±0.005	Г (mm/s) ±0.005	*Qs* (mm/s) ±0.005	HF (kOe) ±0.4	Spectral Area(%)
Sextet	0.033	0.262	−0.082	331.3	96
Singlet	−0.057	0.182	0.000		4

**Table 9 molecules-27-07190-t009:** Extrinsic magnetic properties of the Nd_2_Fe_14_B system. Fe_90_Al_10_ sintered at 300, 400 and 500 °C, respectively.

Temperature (°C)	Hc (Oe) ±0.4	Mr (emu/g) ±0.4	Ms (emu/g) ±0.4
300	171.2	16.5	173.2
400	149.1	12.0	162.9
500	132.2	9.9	138.6

## Data Availability

The data presented in this study were contained within the article.

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
