# Peer review of "Production and Characterization of Nanostructured Powders of Nd2Fe14B and Fe90Al10 by Mechanical Alloying"

_molecules, 2022, doi:10.3390/molecules27217190_

Round 1

Reviewer 1 Report

Dear Authors,

Thank you for your manuscript. It is devoted to an interesting and important area, nevertheless, the overall quality of the manuscript is insufficient, in my opinion. I have some comments on it:

1. The Abstract does not contain any information on motivation of the study or on its novel scientific results.

2. The Introduction section is too short, contains only four literature sources and therefore has to be extended.

3. The goal of the study is not pointed in the Introduction study. What was the reason for the investigations? Which scientific problems were to be solved? 

4. The manuscript style/formatting is not suitable for submission without preliminary fixing. See the appropriate MDPI template.

5. In the Experimental section, separate subsections should describe in detail the methodology for preparing experimental samples, the sources of acquisition of reagents and their quality, as well as the model, manufacturer and country of origin of the scientific equipment used.

6. The manuscript contains typos, e.g.

Line 88, "Ǻ";

Line 165, "remaining magnetization";

Line 192, "Nd2Fe14B";

Line 199, "XRD of the";

Table 3, "67,5";

Table 8, "área";

Line 289, "Nd2Fe14system. Fe90Al10", etc.

7. The TEM images for Nd2Fe14B and Nd2Fe14B/Fe90Al10 systems are needed. The quality of the TEM image for Fe90Al10 system is insufficient. 

8. The resolution and the quality of the figures should be improved.

9. Conclusion section is too short and does not provide any detailed information on the newer results obtained during the study. The surprising result about the disappearance of the hard magnetic properties of the composition of two intermetallic systems should be explained in detail and confirmed by independent methods, in addition to those used in this work. In addition, it is necessary to substantiate the data obtained, based on literary sources, due to the fact that the intermetallic systems studied in the work have been studied in detail for a long time.

Author Response

Response to Reviewer 1 Comments

Point 1: The Abstract does not contain any information on motivation of the study or on its novel scientific results.

Response 1: We have already modified the abstract to include the motivation and the results obtained in the work.

Point 2: The Introduction section is too short, contains only four literature sources and therefore has to be extended.

Response 2: We have already modified the introduction, making it more extensive and including new references.

Point 3: The goal of the study is not pointed in the Introduction study. What was the reason for the investigations? Which scientific problems were to be solved? 

Response 3: We have added to the introduction the main objective of the research, mentioning what was the problem to be solved associated to the results obtained by other authors.

Point 4: The manuscript style/formatting is not suitable for submission without preliminary fixing. See the appropriate MDPI template.

Response 4: We have adjusted the style of the article to the MDPI journal format.

Point 5: In the Experimental section, separate subsections should describe in detail the methodology for preparing experimental samples, the sources of acquisition of reagents and their quality, as well as the model, manufacturer and country of origin of the scientific equipment used.

Response 5: We have rewritten the experimental section and included the methodology used in the preparation of samples.

Point 6: The manuscript contains typos, e.g.

Line 88, "Ǻ";

Line 165, "remaining magnetization";

Line 192, "Nd2Fe14B";

Line 199, "XRD of the";

Table 3, "67,5";

Table 8, "área";

Line 289, "Nd2Fe14B system. Fe90Al10", etc.

Response 6: We have reviewed and modified all the errors mentioned by the referee.

Point 7: The TEM images for Nd2Fe14B and Nd2Fe14B/Fe90Al10 systems are needed. The quality of the TEM image for Fe90Al10 system is insufficient. 

Response 7: We have included a new TEM image for the Fe90Al10 sample. Unfortunately we do not have more financial resources to perform additional microscopy measurements for the Nd2Fe14B and the Nd2Fe14B/Fe90Al10 nanocomposite samples.

Point 8: The resolution and the quality of the figures should be improved.

Response 8: We have improved the quality of all figures (.TIFF).

Point 9: Conclusion section is too short and does not provide any detailed information on the newer results obtained during the study. The surprising result about the disappearance of the hard magnetic properties of the composition of two intermetallic systems should be explained in detail and confirmed by independent methods, in addition to those used in this work. In addition, it is necessary to substantiate the data obtained, based on literary sources, due to the fact that the intermetallic systems studied in the work have been studied in detail for a long time.

Response 9: We have modified the conclusions of the work as suggested by the referee, making them more extensive and correlating the results obtained in this work with the results obtained by other authors.

Reviewer 2 Report

I recommend the following minor revisions:

1. Improve the literature review, including recent references (2021-2022).

2. Improve the presentation of the figures. E.g. Figure 7, the numbers in y axes are shown 1,00 while in Tables with format 1.00. Figure 5. Some numbers in y axes are overlapped.

3. Discuss if the differences in the parameters of Table 1 are significant.

Author Response

Response to Reviewer 2 Comments

Point 1: Improve the literature review, including recent references (2021-2022).

Response 1: We have updated the references as suggested by the referee to recent studies (2020-2022).

Point 2: Improve the presentation of the figures. E.g. Figure 7, the numbers in y axes are shown 1,00 while in Tables with format 1.00. Figure 5. Some numbers in y axes are overlapped.

Response 2: We have improved the quality of the figures (.TIFF), and we have made the changes suggested by the referee.

Point 3: Discuss if the differences in the parameters of Table 1 are significant.

Response 3: The differences in the X-ray parameters are due to the particle size reduction by the surfactant-assisted mechanical alloying process. However, the lattice parameter of the Fe90Al10 alloy does not change with milling time, because the mechanical energy provided by the planetary mill is only able to reduce the grain size due to the brittleness of the FeAl alloy, but it is not enough to modify the lattice parameter of the alloy.

Reviewer 3 Report

1) I advise to revise the style of the manuscript carefully, because there are some hard-to-understand phrases. For example, the Introduction section, lines 52-54 “Subsequently, a mixture of the two sintered systems Nd2Fe14B/Fe90Al10 was magnetically characterized at different temperatures (300°C, 400°C, and 500°C) for one hour”: what did the authors mean by “magnetically characterized at different temperatures”?

2) Figure 1 I suggest providing the XRD card number from the database that was used to identify the phases.

3) Subsection 3.1. “Figure 2 shows the TEM micrograph of the NdFeB nanoflakes…”: as follows from the text above on this stage the authors carried out experiments on Fe90Al10. How did NdFeB nanoflakes appear there? Please, explain.

4) Subsection 3.2, XRD patterns. In my opinion, the theoretical curve (red) for mechanically milled samples (2d and 3d XPD patterns in Figure 6) does not agree well with the commercial one. I suggest improving the quality of the experimental patterns and their theoretical approximations.

5) Please explain the peculiarities of the peak located at 2θ = 26.5 °. It is well defined in commercial powder. After grinding for 24 h, this peak disappears/is of low intensity, but its intensity increases when the powder is milled for 48 h. I also recommend labeling this peak in the figure for the reader's convenience.

6) I strongly recommend to provide SEM images.

Author Response

Response to Reviewer 3 Comments

Point 1: I advise to revise the style of the manuscript carefully, because there are some hard-to-understand phrases. For example, the Introduction section, lines 52-54 “Subsequently, a mixture of the two sintered systems Nd2Fe14B/Fe90Al10 was magnetically characterized at different temperatures (300°C, 400°C, and 500°C) for one hour”: what did the authors mean by “magnetically characterized at different temperatures”?

Response 1: We have reviewed the writing style of the article and have made the modifications suggested by the referee.

Point 2: Figure 1 I suggest providing the XRD card number from the database that was used to identify the phases.

Response 2: We have included the ICSD number of phases used in X-ray refinement.

Point 3: Subsection 3.1. “Figure 2 shows the TEM micrograph of the NdFeB nanoflakes…”: as follows from the text above on this stage the authors carried out experiments on Fe90Al10. How did NdFeB nanoflakes appear there? Please, explain.

Response 3: We have made a typographical error in typing the NdFeB alloy. But we have already modified the text for a better understanding.

Point 4: Subsection 3.2, XRD patterns. In my opinion, the theoretical curve (red) for mechanically milled samples (2d and 3d XPD patterns in Figure 6) does not agree well with the commercial one. I suggest improving the quality of the experimental patterns and their theoretical approximations.

Response 4: According to the X-ray results of the Nd2Fe14B alloy, the structural parameters show a reduction in particle size due to the broadening of the diffraction peaks. This is mainly due to the increase in microtensions promoted by the milling time. As a consequence it contributes to the poor definition of the diffraction peaks and this affects the Rietveld refinement.

Point 5: Please explain the peculiarities of the peak located at 2θ = 26.5 °. It is well defined in commercial powder. After grinding for 24 h, this peak disappears/is of low intensity, but its intensity increases when the powder is milled for 48 h. I also recommend labeling this peak in the figure for the reader's convenience.

Response 5: We have made a magnification around peak located at 2θ = 26.5 ° but we have not observed a broadening of this peak and we note that the dots are along the same position which indicates a bad diffractogram statistic.

Point 6: I strongly recommend to provide SEM images.

Response 6: We have included a new TEM image for the Fe90Al10 sample. Unfortunately we do not have more financial resources to perform additional microscopy measurements (SEM) for the Fe90Al10,  Nd2Fe14B and the Nd2Fe14B/Fe90Al10 nanocomposite samples.

Round 2

Reviewer 1 Report

Dear Authors,

Thank you for your careful improvement of the manuscript. I believe now it can be suitable for publication in the present form.

Reviewer 3 Report

The authors made all the necessary corrections. I am recommending the acceptance of the manuscript.